# Molecular Docking and Dynamics Simulation of Natural Compounds from Betel Leaves (*Piper betle* L.) for Investigating the Potential Inhibition of Alpha-Amylase and Alpha-Glucosidase of Type 2 Diabetes

**DOI:** 10.3390/molecules27144526

**Published:** 2022-07-15

**Authors:** Sabbir Ahmed, Md Chayan Ali, Rumana Akter Ruma, Shafi Mahmud, Gobindo Kumar Paul, Md Abu Saleh, Mohammed Merae Alshahrani, Ahmad J. Obaidullah, Sudhangshu Kumar Biswas, Md Mafizur Rahman, Md Mizanur Rahman, Md Rezuanul Islam

**Affiliations:** 1Department of Biotechnology and Genetic Engineering, Faculty of Biological Sciences, Islamic University, Kushtia 7003, Bangladesh; sabbir.btge@gmail.com (S.A.); chayanali7@gmail.com (M.C.A.); rumabtge.iu@gmail.com (R.A.R.); shu_genetics@yahoo.com (S.K.B.); rahmanmm@btge.iu.ac.bd (M.M.R.); rezuanul@btge.iu.ac.bd (M.R.I.); 2Division of Genome Sciences and Cancer, The John Curtin School of Medical Research and The Shine-Dalgarno Centre for RNA Innovation, The Australian National University, Canberra, ACT 2601, Australia; shafi.mahmud@anu.edu.au; 3Department of Genetic Engineering and Biotechnology, University of Rajshahi, Rajshahi 6205, Bangladesh; gobindokumar38@gmail.com (G.K.P.); saleh@ru.ac.bd (M.A.S.); 4Department of Clinical Laboratory Sciences, Faculty of Applied Medical Sciences, Najran University, Najran 61441, Saudi Arabia; mmalshahrani@nu.edu.sa; 5Drug Exploration and Development Chair (DEDC), Department of Pharmaceutical Chemistry, College of Pharmacy, King Saud University, Riyadh 11451, Saudi Arabia; aobaidullah@ksu.edu.sa; 6Department of Pharmaceutical Chemistry, College of Pharmacy, King Saud University, Riyadh 11451, Saudi Arabia

**Keywords:** apigenin-7-*O*-glucoside, antidiabetic drugs, in silico analysis, *Piper betle* L., type 2 diabetes

## Abstract

*Piper betle* L. is widely distributed and commonly used medicinally important herb. It can also be used as a medication for type 2 diabetes patients. In this study, compounds of *P. betle* were screened to investigate the inhibitory action of alpha-amylase and alpha-glucosidase against type 2 diabetes through molecular docking, molecular dynamics simulation, and ADMET (absorption, distribution, metabolism, excretion, and toxicity) analysis. The molecule apigenin-7-*O*-glucoside showed the highest binding affinity among 123 (one hundred twenty-three) tested compounds. This compound simultaneously bound with the two-target proteins alpha-amylase and alpha-glucosidase, with high molecular mechanics-generalized born surface area (MM/GBSA) values (ΔG Bind = −45.02 kcal mol^−1^ for alpha-amylase and −38.288 for alpha-glucosidase) compared with control inhibitor acarbose, which had binding affinities of −36.796 kcal mol^−1^ for alpha-amylase and −29.622 kcal mol^−1^ for alpha-glucosidase. The apigenin-7-*O*-glucoside was revealed to be the most stable molecule with the highest binding free energy through molecular dynamics simulation, indicating that it could compete with the inhibitors’ native ligand. Based on ADMET analysis, this phytochemical exhibited a wide range of physicochemical, pharmacokinetic, and drug-like qualities and had no significant side effects, making them prospective drug candidates for type 2 diabetes. Additional in vitro, in vivo, and clinical investigations are needed to determine the precise efficacy of drugs.

## 1. Introduction

Type 2 diabetes mellitus (DM) is a lifelong disorder and rising resolutely around the world [1,2]. People with type 2 DM are more vulnerable to many difficulties and have increased odds of common infection [3], thereby easily creating a variety of complex diseases such as kidney damage, dysfunction of the brain, and cancer, resulting in increased morbidity and mortality [4]. This most common type of chronic disease is denoted by hyperglycemia and insulin resistance [5]. Hyperglycemia occurs when the blood glucose level rises due to a lack of insulin from the pancreatic cell [6]. One of the therapeutic ways to regulate postprandial hyperglycemia in type 2 DM is by inhibiting the metabolism of dietary carbohydrates [7]. First, dietary carbohydrates break down into monosaccharides by alpha-amylase activity in the digestive system. This monosaccharide is then converted to glucose by alpha-glucosidase and driven to the bloodstream on absorption [8]. Therefore, inhibiting alpha-amylase and alpha-glucosidase enzyme activity can reduce carbohydrate metabolism, therefore decreasing glucose levels in the blood [9].

Various types of synthetic medicines have already been developed, such as oral hypoglycemic drugs such as metformin, sulfonylureas, thiazolidinediones [10], biguanides, meglitinides, dipeptidyl peptidase-IV inhibitors [11], etc., to treat type 2 DM, but long-term use of these therapeutic agents may create severe side effects including liver complexity, hypoglycemia, diarrhea, and others [12]. It is now important to develop new inhibitors with high efficacy and low toxicity. In this circumstance, herbal medicinal plants can be a better source for manufacturing new drugs because of their lower toxicity, specificity, target affinity, and plentiful nature [13]. Various plants, including *Vinca rosea* (Nayantara), *Allium sativum* (Garlic), *Trigonella foenum* (Fenugrec), etc., with their chemical compounds including phenolic, flavonoids, and others, have been shown to have effective medicinal values to decrease the level of glucose in the blood [14].

Despite extensive studies on medicinal herbs to scientifically establish their potential medicinal properties, there are a lot of latent possible healing properties in their ethnomedicinal applications [15]. Herein, the medicinal herb *Piper betle* is not an anomaly. *P. betle* belongs to the genus piper and family Piperaceae, and is a dioecious, annual creeper, almost one meter long, usually grown in South East Asian countries such as India, China, and Vietnam [16]. *P. betle* leaves are mostly used for many medicinal purposes because of their potent pharmaceutical properties [17]. Several phytochemical analyses have been undertaken on *P. betle* by scholars who stated that it contains a large number of bioactive molecules which act as anti-diabetic agents [18]. It was first reported that the oral administration of betel leaves extracts in fasted normoglycemic & streptozotocin-induced diabetic rats significantly decreased sugar levels in the blood [19]. Later, Srividya et al. found that hydroxychavicol from *P. betle* leaves extract has antidiabetic activity [20]. Moreover, the whole plant extract of *P. betle* significantly reduced the glucose levels in the blood, which was demonstrated on healthy rats by Willer et al. [21]. A further study noted that the ethanolic extract of *P. betle* leaves showed potent inhibition properties against alpha-amylase [22]. Betel leaves extract also has several medicinal values, such as possessing anti-inflammatory, anti-depressant, antioxidant, antibacterial, anti-infective, anti-cancer, anti-asthmatic, platelet inhibitor etc., properties [18]. However, none of the research studies have been performed to discover drugs from *P. betle* compounds against type 2 DM as alpha-amylase and alpha-glucosidase inhibitors using computational approaches. This research study was designed to reveal the specific interaction between two or more molecules (molecular recognition) and the binding mechanism of betel compound to alpha-amylase and alpha-glucosidase by utilizing molecular modelling and molecular dynamics simulation studies [13]. The alpha-amylase enzyme contains catalytic residues such as ASP-197, GLU-233, & ASP-300 which act on both α-D-(1,4) linkages and α-D-(1,6) linkages of large oligosaccharide molecules and break them into disaccharides and trisaccharides [8].These disaccharides and trisaccharides further break down into monomer molecules with the help of alpha-glucosidase which contains ASN-258, 327, 382 and ILE-143 catalytic residues [23].

In the starch digesting mechanism, alpha-amylase and -glucosidase are promising targets for treating Type 2 DM [24]. To discover potent inhibitors from betel plants, 123 phytochemicals have been screened virtually with both enzymes, alpha-amylase and alpha-glucosidase by applying in silico approaches. The virtual screening protocol was conducted using the three-stage filtering technique, (1) docking with Glide XP and ADME/T analysis, (2) using MM-GBSA analysis, and (3) the Prime MM-GBSA module was operated to obtain a more reliable estimation. Thereafter, the flexibile 50-ns MD simulations were run on docked complexes with high MM-GBSA values to illustrate the binding pose of target compounds. Therefore, the main finding of this study was to investigate the alpha-amylase and alpha-glucosidase inhibitors from betel leaves’ compounds through computational approaches, with the hypothesis that the inhibitors could ultimately be used as drug candidates for treating Type 2 DM.

## 2. Results

### 2.1. Virtual Screening Based on Docking Scores of Compounds from Betel Leaves (Piper betle L.)

Virtual screening is a useful technique for molecular docking which identifies lead compounds in drug discovery. Here, we conducted a molecular docking analysis of previously identified (a total of 123) compounds with alpha-amylase and alpha-glucosidase. In the case of alpha-amylase, the top (*n* = 7) compounds were chosen based on the best docking scores using the Glide XP docking program. We considered the cut-off value to be 6. Therefore, the best docking score for luteolin-7-*O*-glucoside was −8.504 kcal mol^−1^, while the lowest docking score for ellagic acid was −6.164 kcal mol^−1^, whereas the docking scores for the control inhibitors, acarbose was −9.298. On the other hand, seven compounds were selected as alpha-glucosidase inhibitors, and their cut-off value was considered to be 7. The highest and lowest docking scores were −12.094 and −7.317 kcal mol*^−^*^1^ for luteolin-7-*O*-glucoside and chlorogenic acid, respectively. Here the control inhibitor docking score was −13.142 kcal mol*^−^*^1^.

### 2.2. MM-GBSA Binding Affinity Estimation

To obtain a more reliable estimation, the selected seven (*n* = 7) compounds were subjected to MM-GBSA analysis based on their binding affinity calculation. Among them, the best three ligands with the highest binding affinity (ΔG Bind) with the receptor’s catalytic pair were selected for both alpha-amylase and alpha-glucosidase. Here, for alpha-amylase, the maximum binding affinity was found in apigenin-7-*O*-glucoside (ΔG Bind = −45.02 kcal mol*^−^*^1^), while the remaining two compounds, luteolin-7-*O*-glucoside and quercetin, showed a binding affinity (ΔG Bind = −35.602 kcal mol*^−^*^1)^ and (ΔG Bind = -34.664 kcal mol*^−^*^1^) respectively. On the other hand, for alpha-glucosidase, the maximum binding affinity was found in apigenin-7-*O*-glucoside −38.28 kcal mol*^−^*^1^. The other two compounds had the binding affinity (ΔG Bind = −34.664kcal mol*^−^*^1^) and (ΔG Bind = −28.688 kcal mol*^−^*^1^), respectively. In parallel to the ligands that have been identified, the control inhibitor acarbose had binding affinities of −36.796 kcal mol*^−^*^1^ for alpha-amylase and −29.622 kcal mol*^−^*^1^ for alpha-glucosidase, respectively. The highest MM-GBSA assay values were considered for the next steps compared to the control, acarbose. In addition to this, before docking with the *P. betle* compounds, we redocked the established and native ligand acarbose with alpha-amylase and alpha-glucosidase (Appendix A). The RMSD difference was approximately 2.25 Å (for alpha-amylase) and 2.6 Å (for alpha-glucosidase), which is lower than the allowed value of 3.0 Å.

### 2.3. Ligand Binding Analysis

BIOVIA Discovery studio visualizer v 4.5 (BIOVIA, San Diego, CA, USA) was used to visualize the molecular interactions of the chosen compounds [25]. Here, we presented molecular interactions with both enzymes with the best two compounds, (apigenin-7-*O*-glucoside and Luteolin-7-*O*-glucoside). The best compound apigenin-7-*O*-glucoside binds with the active sites of alpha-amylase residues, ASP-300, GLU-233, ASP-197, GLY-306, HIS-305, and GLN-63 single hydrogen bonds, while ASP-197 created double hydrogen bonds, with bonding distances of 1.86 and 1.59 Å, respectively. Another hydrogen bond was shown in ASP-300. However, TRP-59 developed substantial hydrophobic interactions with the ligand via Pi-Pi stacked bonding and additional unfavourable donor-donor interactions were found in ARG-195 with the ligand (Figure 1a and Table 1).

Similarly, apigenin-7-*O*-glucoside formed several hydrogen bonds with the alpha-glucosidase active site residues ASP-60, ASN-258, ASP-327, ILE-143, and ASP-382, where all residues except ASP-60, ASN-258 generated double hydrogen bonds with the ligand. Besides this, another three hydrogen bonds were shown in ARG-411, GLY-384, and GLY-410 residues with bonding distances of 2.05, 3.09, and 2.67 Å, respectively. However, PHE-163 and TYR-63 were responsible for hydrophobic interaction through Pi-Pi stacked and Pi-Pi T-shaped bonding. Furthermore, an electrostatic interaction appeared in ASP-327 residues (Figure 2a and Table 2).

The control compound acarbose binds with the alpha-amylase active site residues GLU-240, GLY-306, HIS-305, ASP-197, ASP-300, THR-163 through conventional hydrogen bond and GLY-306, ASP-300 by carbon hydrogen bond (Figure 1b and Table 1). Similarly, Alpha-glucosidase contains catalytic residues such as ASN-258, ASP-327, ILE-143, and ASP-382 acarbose bound with them through a conventional hydrogen bond (Figure 2b and Table 2).

We described the second most suitable compound, luteolin-7-*O*-glucoside, based on the MM-GBSA value (−35.602 kcal mol^−1^ for alpha-amylase and −34.664 kcal mol^−1^ for alpha-glucosidase), which is merely similar to control acarbose (−36.796 kcal mol^−1^ for alpha-amylase and −29.622 kcal mol^−1^ for alpha-glucosidase).

Luteolin-7-*O*-glucoside and alpha-amylase active site residues with ASP-300, GLU-233, HIS-299, ASP-356, ARG-195 and GLN-63 whereas ASP-356 and GLN-63 created double and triple conventional hydrogen Bond with the ligand (Figure 3a and Table 1).

Similarly, luteolin-7-*O*-glucoside formed several hydrogen bonds with the alpha-glucosidase active site residues with HIS-103, ASP-60, ILE-143, ASP-382, THR-409, ASN-258, and ARG-411 producing the most interactions. However, carbon hydrogen interactions with the ligand were also identified in ASP-327, GLY-410, and GLY-384 residues. ASP-199 also formed an electrostatic bond through Pi-anion with a bonding distance of 4.32 Å. Hydrophobic interactions were also found in the active site residues of ALA-200, PHE-144, and PHE-163 (Figure 3b and Table 2). Earlier we mentioned the control compound acarbose with the alpha-amylase and alpha-glucosidase active sites catalytic residues in Table 1; Table 2.

For a suitable drug candidate selection, we need to perform the chemical absorption, distribution, metabolism, excretion, and toxicity (ADMET) analysis using SwissADME and QikProp. These tools predict a wide range of chemical and physical properties of a drug candidate.

### 2.4. ADMET Analysis Values

Drug development is primarily concerned with bioavailability and toxicity. Hence, such compounds were subjected to ADME/T profiling. The ADMET is a machine learning in silico analysis software that predicts the compounds-based absorption, distribution, metabolism, excretion, and toxicity parameters. Detailed information on ADMET results is provided in Table 3.

### 2.5. Molecular Dynamics Simulation

The molecular dynamics simulation study was conducted to understand the flexibility level of the complexes as well as the apo systems. The root mean square deviations of the C-alpha atoms of the complexes were calculated to explore the rigidity of the complexes. The apo alpha-amylase, apigenin-7-*O*-glucoside, and acarbose complexes had an initial upper trend in RMSD due to a higher level of flexibility. Similar trends were observed for the glucosidase complexes. The apo alpha-amylase reached a stable state after 20 ns, whereas the apo alpha-glucosidase also stabilized after 30ns. In both systems, the complexes had some degree of deviations for the rest of the period, but they maintained the stable state in whole simulation times, and also the RMSD profile from both systems was below 2.5 Å, which defines the stable state of the complexes (Figure 4 and Figure 5).

The solvent-accessible surface area of the complexes was analyzed to understand the changes in the protein surface area, where higher SASA related to the expansion of the surface area and the lower SASA related to the truncation of the protein volume. The apo alpha-amylase had higher SASA than the docked complexes (apigenin-7-*O*-glucoside and acarbose) across the simulation times which indicate the alpha-amylase had rigid binding affinity with the ligand molecules. The complexes reached in a stable state after 40ns and maintained rigidity for the rest of the simulation times (Figure 4c). The alpha- glucosidase complexes had a similar trend, as their complexes also reduced the SASA profile upon the bindings with the ligand molecules (Figure 5c).

The radius of the gyration profile of the simulations systems was also calculated to examine the liability of the complexes. The Rg trend of the amylase systems followed the steady-state and did not change too much, which defines the lower mobile nature of the complexes (Figure 4d). Therefore, the apo alpha-amylase and apigenin-7-*O*-glucoside complexes from glucosidase had a similar Rg value, whereas the acarbose from glucosidase had lower Rg, which provide information of compactness of the enzyme-ligand complexes (Figure 5d). Furthermore the hydrogen bond pattern of the protein systems defines the stable trend. The figure indicates that the complexes from the alpha-amylase and alpha-glucosidase complexes had a stable trend in the hydrogen bond across the simulation times (Figure 4e and Figure 5e).

Moreover, root mean square deviations of the complexes were analyzed to understand the flexibility level across the amino acid residues of the complexes. The RMSF from the alpha-amylase complexes had a value below 2.5 Å except Trp134, Thr155, and Asn350. The alpha-glucosidase complexes also had a lower RMSF profile for maximum residues, which defines the stability level of these complexes (Figure 4b and Figure 5b).

### 2.6. Pharmacokinetics and Drug Likeliness Properties

The toxicity level, drug-likeliness properties, pharmacokinetics, and physicochemical properties of the promising drug compound (apigenin-7-*O*-glucoside) from *P. betle* are mentioned in Table 3 and Appendix A. The different parameters are important for selection of an ideal drug candidate. Specifically, physicochemical properties are important for a reliable drug design.

The total polar surface area (TPSA) value of apigenin-7-*O*-glucoside was 170.05 angstroms squared (Å2), while the TPSA value of control acarbose was 321.17 Å^2^. The molecules with a low polar surface area are good at permeating cell membranes. The physicochemical properties of apigenin 7-glucoside obeyed the Lipinski’s rule of five with one violation (NH or OH > 5), and this compound had a bioavailability score (0.55) that indicates good oral adsorption. Furthermore, the consensus Log Po/w (the average of all five lipophilicity predictions) of this compound is less than 5, which is within the acceptable range; the drug score and drug-likeliness indicate that this compound is more appropriate to be used as a drug. The Log Po/w value of compound apigenin-7-*O*-glucoside is 2.17 (Appendix A). Few parameters are good and few are similar to the control acarbose (Appendix A). Based on physicochemical properties, lipophilicity, water-solubility, pharmacokinetics properties, drug-likeliness activity, toxicity levels, and other parameters suggest that the apigenin-7-*O*-glucoside compound could be used as a drug candidate for DM patients.

## 3. Discussion

A number of *P. betle* compounds had proved for an alternative source of medication of human diseases through several research studies [26,27,28]. This research aimed to investigate the inhibitory action of alpha-amylase and alpha-glucosidase by predicting the binding affinity among the *P. betle* phytochemicals, applying in silico molecular docking studies and molecular dynamic simulation. Two enzymes, alpha-amylase and alpha-glucosidase, are responsible for the carbohydrates’ breakdown [29]. Therefore, inhibitions of these enzymes may cause carbohydrate digestion to be delayed and glucose absorption to be reduced; as a result, the blood glucose increase after a meal is reduced [9]. Acarbose is an alpha-amylase inhibitor that aids in lowering postprandial hyperglycemia in clinical trials, but this could increase the risk of cardiovascular disease in people with no cardiovascular issues [30]. Furthermore, acarbose treatment for an extended period decreases overall cholesterol and triglycerides in blood patients affected by diabetes [31]. Furthermore, alpha-amylase and alpha-glucosidase are inhibited by drugs like acarbose, voglibose, and miglitol, but they create some unfavourable side effects in the body like bloating, intestinal discomfort, diarrhea, etc. [32].

In consideration of this problem, various herbal medications have been used to treat diabetes. Consequently, several natural herbal plants are already being used to treat a variety of metabolic diseases such as DM. The ethnopharmacological study lists over 1200 plants with anti-diabetic activity that are used to treat DM, demonstrating the significance of conventional folk medicines [33]. Among them, previous reports in several plants showed inhibitory effects on alpha-amylase and alpha-glucosidase enzymes [29,34,35]. Jia et al. [1] showed that apigenin-7-*O*-glucoside showed the highest (IC_50_ = 22.80 ± 0.24 μM, compared to control chemical acarbose) inhibition activity against alpha-glucosidase enzyme among the tested 27 dietary flavonoid compounds [1]. Similarly, in this study, the apigenin-7-*O*-glucoside compound showed the highest binding affinity among 123 (one hundred twenty-three) tested compounds through molecular docking, molecular dynamics simulation, and ADMET, and ultimately it could be used as an effective alpha-glucosidase inhibitor and is a promising target drug candidate for treating Type 2 DM.

The alpha-amylase structure was detected by the X-ray diffraction method. It has 496 amino acids and is composed of three domains, Domain A, Domain B, and Domain C. Between Domains A and B, the residues of active sites ASP-197, GLU-233, and ASP-300 were found. Further investigation found that the catalytic activity of the enzyme is reduced by 1000× when the side chains of GLU-233 or ASP-300 are replaced. The ASP-300 variant of alpha-amylase from the human was structurally analyzed and its complex with acarbose revealed its significance in the binding mode of the inhibitor [34].

According to our findings, our top candidates bind with the catalytic residues of both targets, alpha-amylase and alpha-glucosidase. Consequently, apigenin-7-*O*-glucoside was stabilized by several hydrogen and hydrophobic bonds. As can be seen in Table 1, hydrogen bonds were found in with catalytic residues ASP-300, GLU-233, and ASP-197, while the ligand and ASP-197 created double hydrogen bonds, with bonding distances of 1.86 and 1.59 Å, respectively, of the alpha-amylase. In a study [4], antidiabetic activities and molecular docking analysis were assessed with alpha-amylase and alpha-glucosidase enzyme inhibition from plant compounds (3-oxolupenal and katononic). The molecular docking analysis confirmed the inhibitory action of 3-oxolupenal and katononic acid by the screened compounds. The three amino acid residues (ASP-197, GLU-233, and ASP-300) are important for the active site of alpha-amylase through X-ray structure and enzyme kinetics [35]. In addition, ASP-197 acts as a nucleophile, GLU-233 acts as a starch hydrolysis reaction, and ASP-300 acts as the orientation of the substrate [4]. Similarly, we observed the three amino acid residues, such as ASP-197, GLU-233, and ASP-300 (Table 1, Figure 1a), with alpha-amylase-apigenin-7-*O*-glucoside complex, which are important for catalysis and docking pose orientation similar to control acarbose X-ray structure and docking pose orientation (Appendix A).

Alpha-glucosidase contains catalytic residues such as ASN-258, ASP-327, ILE-143, and ASP-382 [23]. Similarly, apigenin-7-*O*-glucoside formed several hydrogen bonds with the active site residues ASP-60, ASN-258, ASP-327, ILE-143, and ASP-382, where all residues except ASP-60 and ASN-258 generated a double hydrogen bond with the alpha-glucosidase. Besides this, another three hydrogen bonds were shown in ARG-411, GLY-384, and GLY-410 residues with bonding distances of 2.05, 3.09, and 2.67 Å, respectively (Table 2). However, PHE-163 and TYR-63 were responsible for hydrophobic interaction through Pi-Pi stacked and Pi-Pi T-shaped bonding. Furthermore, an electrostatic interaction was present in ASP-327 residues located in the active site of enzyme.

The molecular mechanics/generalized Born surface area (MM/GBSA) approach for calculating protein-ligand binding free energies has become popular due to its excellent mix of computational precision and efficiency. In structure-based drug design, it is extensively used to predict end-point binding free energy [36]. MM/GBSA analysis is a much more precise prediction and produces better outcomes than most molecular docking scoring [37]. In this study, for alpha-amylase, the maximum binding affinity was found in apigenin-7-*O*-glucoside (ΔG Bind = −45.02 kcal mol^−1^) and for alpha-glucosidase, the maximum binding affinity was (ΔG Bind = −38.288 kcal mol^−1^). Both alpha-amylase and alpha-glucosidase MM/GBSA values of apigenin-7-*O*-glucoside are higher than control inhibitor acarbose binding affinities of −36.796 kcal mol^−1^ and −29.622 kcal mol^−1^ for alpha-amylase, alpha-glucosidase, respectively.

Pharmacokinetic and toxicological properties provide crucial information on the functioning of drug molecules inside the human body [38]. In this research article, the apigenin-7-*O*-glucoside alpha-amylase and alpha-glucosidase complexes’ pharmacokinetic and toxicological properties were calculated from ADMET analysis using SwissADME and QikProp. The tools predict a wide range of chemical and physical properties of drug candidate compounds rapidly and accurately, such as bind molecular weight (MW), SASA, hydrogen bond (HB) donor-acceptor, molecular lipophilicity (Qplog Po/w), QPlogS, Qplog HERG, and human oral absorption rate [39]. The vast majority of drugs on the marketplace have a molecular mass between 200 to 600 Daltons, mostly <500 [40]. Hydrogen (H)-bonds play a key role in protein folding, ligand-protein interaction, and catalysis in biological systems [41]. Similarly, Qplog Po/w, QPlogS, Qplog HERG, and Human oral absorption are the major parameters for the assessment of chemical compounds and for identifying the pharmacokinetic features of the drugs. Here, apigenin-7-*O*-glucoside has satisfied the majority of the requirements, specifically, Lipinski’s Rule of Five, ensuring its drug-likeliness behaviour (i.e., AMES test, Veber rule with no violations, better bioavailability value, and other parameters) and ultimately apigenin-7-*O*-glucoside could be used as a drug candidate (Table 2 and Appendix A).

Molecular dynamics (MD) simulation is an effective technique to understand the stability and dynamics of the protein-ligand complex [42]. MD simulations for drug design reveal the structural voids needed to create new compounds with superior target affinity. The use of MD simulations in drug design can obtain structural data as well as the effect of protein structure stability on ligand binding, resulting in a better sampling of binding poses and more accurate affinity estimates with improved structural accuracy [43]. The greater RMSD, RMSF, Rg, and SASA scores represent the greater flexibility of the system [39]. We conduct molecular dynamics simulations of the docking complexes to study the changes in a protein structure when the ligand binds with it. After 20 ns, the amylase complexes had stabilized, while the glucosidase complex had stabilized after 30 ns. RMSD profiles from both systems were less than 2.5, indicating that the complexes were in a stable condition. On the other hand, SASA, Rg, Hb, & RMSF values (from Figure 4; Figure 5) also denote the complexes’ stability. These computer calculations and statistics might provide crucial information for developing reasonable medication candidates to treat type 2 diabetes. However, therapeutic efficacy and a new drug releases depend on pharmacokinetic properties, efficacy, and safety levels [44,45]. Molecular docking analysis, specifically ADMET analysis of phytochemical (*P. betle* compounds) exhibited a wide range of physicochemical, pharmacokinetic, and drug-likeliness properties, thus in-silico screening is an alternative way for screening new drugs [46] from natural sources [44]. Finally, the compound apigenin-7-*O*-glucoside was exposed to the most stable compound with the highest binding free energy through molecular dynamics simulation.

## 4. Materials and Methods

### 4.1. Ligand Collection and Preparation

In an earlier stage of this investigation, we collected betel leaf compounds from related research findings and literature in the PubMed, Google Scholar, Web of Science, and Scopus databases. Consequently, we developed a dataset of the effective compounds of *P. betle* from PubChem databases, and the structures of ligands of *P. betle* compounds were taken and prepared by using the Ligand preparation wizard of Maestro 11.1 [47] with an OPLS_3 force field for induced-fit docking (IFD) analysis.

### 4.2. Receptor Preparation

The crystal form of human alpha-amylase and the alpha-glucosidase domain were downloaded from the protein data bank archive (PDB ID: 3BAJ and 3W37) [48] and the structure was refined by eliminating water atoms and optimizing protein at neutral pH. To make or set up the ultimate receptor, the GROMACS 96 43B1 algorithm in SWISS-PDB viewer [49] and Chimera (Amber Force field) [50] were used. The reformation was performed for several thiols and hydroxyl groups, amide groups of asparagine, glutamines, and the imidazole ring of histidines, the protonation state of histidines, aspartic acids, and glutamic acids. By using the OPLS_2005 force field [51] by adjusting the maximum heavy atom RMSD to 0.30, Å minimization was completed.

The schematic diagram of this manuscript is presented in Figure 6 as below.

### 4.3. Molecular Docking

The glide module of Schrodinger-Maestro v 9.4 (Schrödinger, LLC, New York, NY, USA) was used for molecular docking of all compounds [52]. To obtain exact compounds with proper biological activity, virtual screening was performed [53]. Here, ligands and receptors are considered flexible & rigid for the period of docking, respectively. Effective compounds were considered based on RMSD (root mean square deviation) standard. Those ligands which have the most negative docking scores and lower (<1 Å) RMSD values were considered for further study. An RMSD value less than 1 angstrom provides information about the confirmation of reference and target protein. Lower value of RMSD between two proteins provides accuracy with docking orientation. The Biovia discovery studio visualizer [25] was used to visualise ligands and receptors’ molecular interaction.

### 4.4. Prime Molecular Mechanics—Generalized Born and Surface Area (MM-GBSA)

The Schrodinger suite’s Prime MM-GBSA module [54] was operated to calculate binding affinity, where a higher negative affinity indicates a higher level of stability. For calculation, a Glide XP docking file with docked Pose viewer was used; Generalized Born Surface Accessible (GBSA) was used as a continuum model in the sample minimization protocol, and molecular mechanics (MM) using the OPLS force field (2005), maintaining the protein’s flexibility [55] VSGB 2.0 [56] was used as a dielectric solvent model for fixing the interactions of π-stacking and H-bond empirical functions.
ΔGbind = Gcomplex − (Gprotein + Gligand), where G = EMM + GSGB + GNP

### 4.5. ADME/T Analysis and Pharmacokinetic and Drug-Likeliness Predictions

QikProp [57] and SwissADME [58] were used to portend the physicochemical properties of ligands, whereas the toxicity test was measured by using pkCSM [59]. The parameters for screening the compounds included the oral absorption rate, solvent available surface area (SASA), QlogP, Lipinski’s rule of five, Veber rule, AMES toxicity, Max. tolerated dose (human), hERG I inhibitor, Hepatotoxicity Skin Sensitization, etc.

### 4.6. Molecular Dynamics Simulations

The AMBER14 force field [60] and the YASARA Dynamics program [61] were used to run MD simulations. The parameters of every ligand in every protein-ligand complex were assigned via Auto SMILE [62] algorithms. As a consequence, unidentified organic molecules are entirely parameterized automatically by calculating semi-empirical AM1 Mulliken point charges [62] using the COSMO solvation model, AM1BCC assigning [63] atom and bond types, as well as allocating GAFF (General AMBER Fore Field) [64] atom types and the remaining parameters of the force field. In a simulation cell, a TIP3P [65] water model was used to optimize and solve the protein-ligand complex’s hydrogen bonding network. This was done before the simulation. At a solvent density of 0.997 gL^−1^, periodic boundary conditions were retained. During solvation, the pKa calculation was performed based on titratable amino acids existing in the protein complex. A method of simulated annealing that utilizes the steepest gradient strategy (5000 cycles) was used to execute the initial energy minimization procedure of each simulation scheme, which is composed of 62,521 ± 10 atoms. At multiple time-step algorithms [60], every simulation was carried out under a physiological environment (298 K, pH 7.4, 0.9 per cent NaCl) [60], with a timestep gap of 2.50 fs. The LINCS (linear constraint solver) [66] algorithm was utilized to restrict all bond lengths, and for water molecules ETTLE was used [67]. The PME [68] approaches were used to explain long-range electrostatic attraction. Ultimately, The MD simulation was completed in 100 nanoseconds with the constant pressure of the Berendsen thermostat [66].

The binding free energy of all snapshots was subjected to MM-PBSA (MMPoisson–Boltzmann surface area) using the YASARA software’s formula.
Binding Energy = EpotRecept + EsolvRecept + EpotLigand + EsolvLigand − EpotComplex − EsolvComplex

## 5. Conclusions

In this study, we explored a new molecule, apigenin-7-*O*-glucoside (screened out from 123 compounds of *Piper betle* L.), which inhibited both enzymes (alpha-amylase and alpha-glucosidase) activity by binding with its active sites, ASP-197, GLU-233, and ASP-300 and ASN-258, ASP-327, ILE-143, ASP-382, respectively. A molecular docking study showed the detailed binding mode of the selected virtual hit with significant bond interactions such as hydrogen, van der Waals, and alkyl bonds etc. It exhibited better MM-GBSA values for both receptors than the control inhibitor molecule, acarbose. According to molecular dynamics and simulation studies, this compound formed stable complexes with receptor proteins owing to different conformational changes. Moreover, ADMET analysis was performed to assess the drug-likeliness proficiency, which demonstrates that it matched Lipinski’s rule of five and Veber’s rules with low or no toxicity. Furthermore, the phytochemical (apigenin-7-*O*-glucoside) showed negative result in the AMES test (presence of mutagenic agents), hepatotoxicity, and the blood-brain barrier test (BBB) for toxicity and drug testing; thus, they could be exploited as drug candidates against type 2 diabetes. Hence, more testing and validation in the wet lab are required to develop an effective and better drug with this phytochemical for type 2 DM treatment.

## Figures and Tables

**Figure 1 molecules-27-04526-f001:**
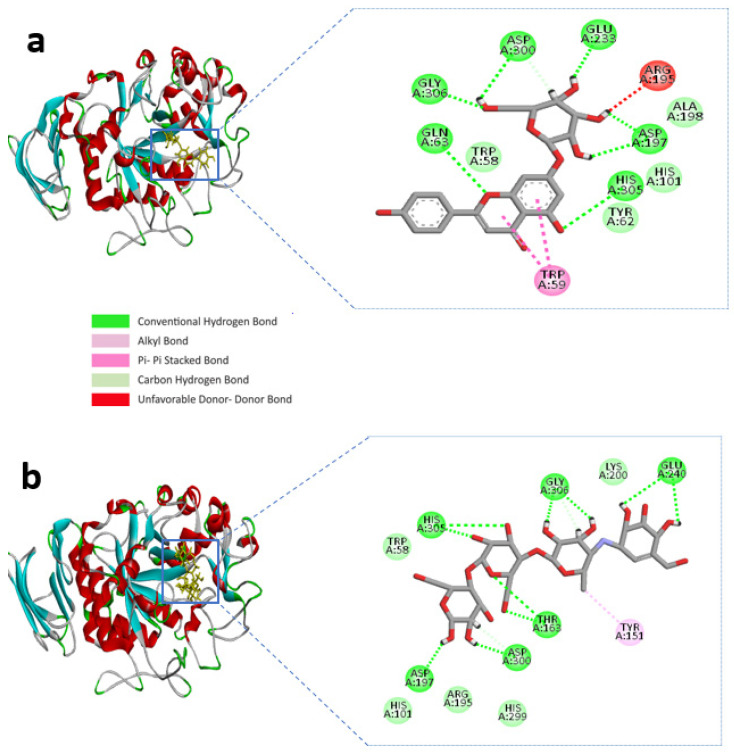
Molecular interactions of the selected compound with alpha-amylase, (**a**) apigenin-7-*O*-glucoside with alpha-amylase, (**b**) control acarbose with alpha-amylase.

**Figure 2 molecules-27-04526-f002:**
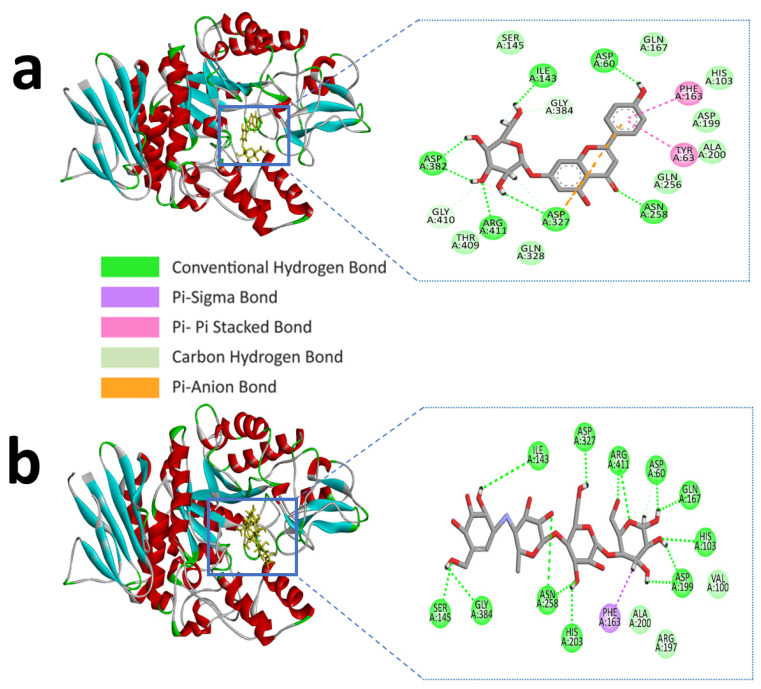
Molecular interactions of the selected compound with alpha-glucosidase, (**a**) apigenin-7-*O*-glucoside with alpha-glucosidase, (**b**) control acarbose with alpha-glucosidase.

**Figure 3 molecules-27-04526-f003:**
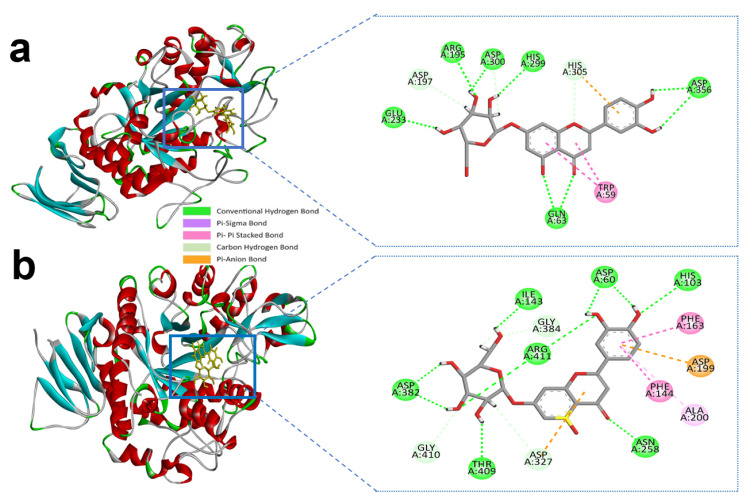
Molecular interactions of selected compound luteolin-7-*O*-glucoside, (**a**) luteolin-7-*O*-glucoside with alpha-amylase, (**b**) luteolin-7-*O*-glucoside with alpha-glucosidase.

**Figure 4 molecules-27-04526-f004:**
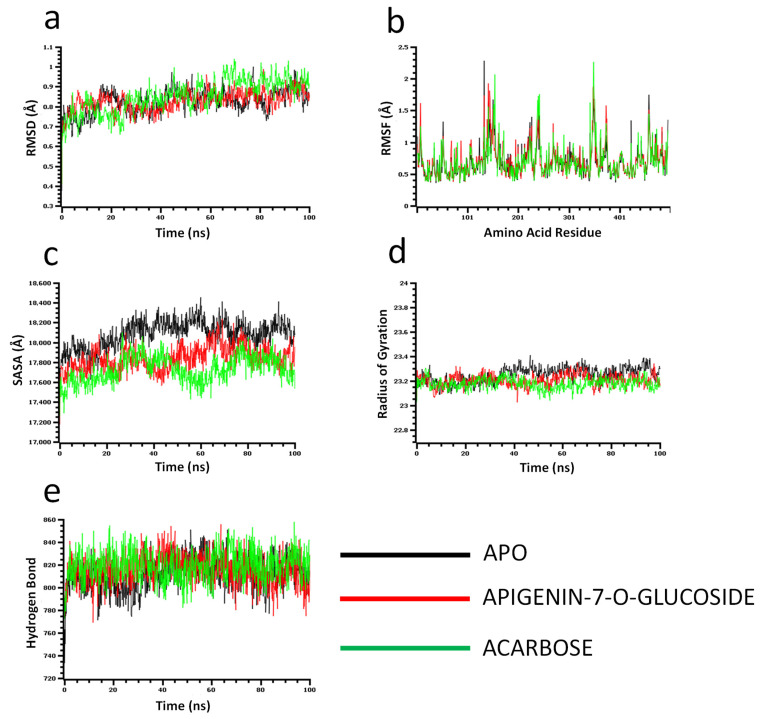
Molecular dynamics simulation for alpha-amylase. Analysis of (**a**) RMSD (Root Mean Square Deviation); (**b**) RMSF (Root Mean Square Fluctuations); (**c**) SASA (Solvent Accessible Surface Area) (**d**) Rg (Radius of Gyration); and (**e**) Binding free energy.

**Figure 5 molecules-27-04526-f005:**
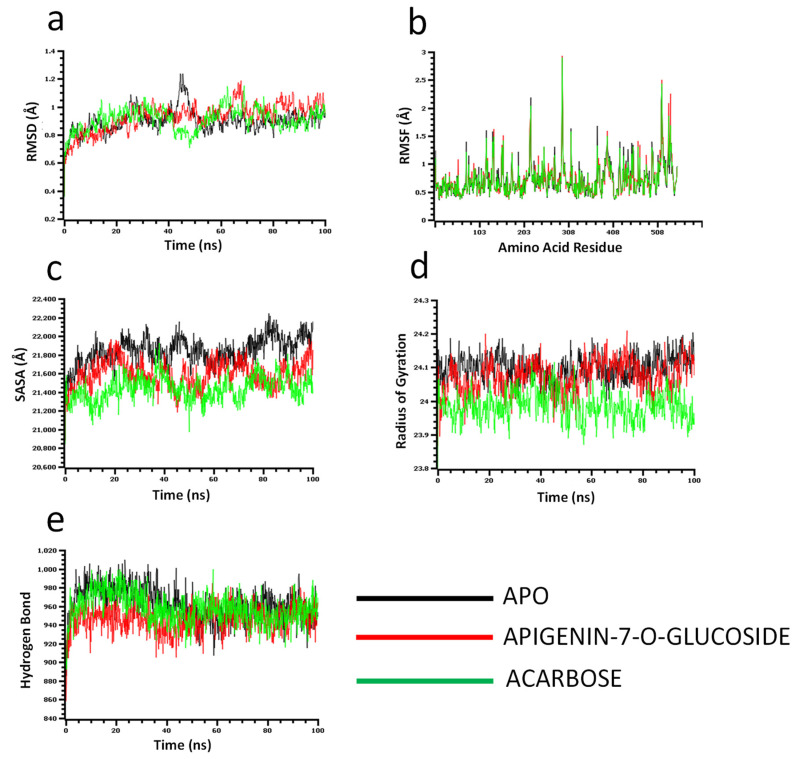
Molecular dynamics simulation for alpha-glucosidase. Analysis of (**a**) RMSD (Root Mean Square Deviation); (**b**) RMSF (Root Mean Square Fluctuations); (**c**) SASA (Solvent Accessible Surface Area) (**d**) Rg (Radius of Gyration); and (**e**) Binding free energy.

**Figure 6 molecules-27-04526-f006:**
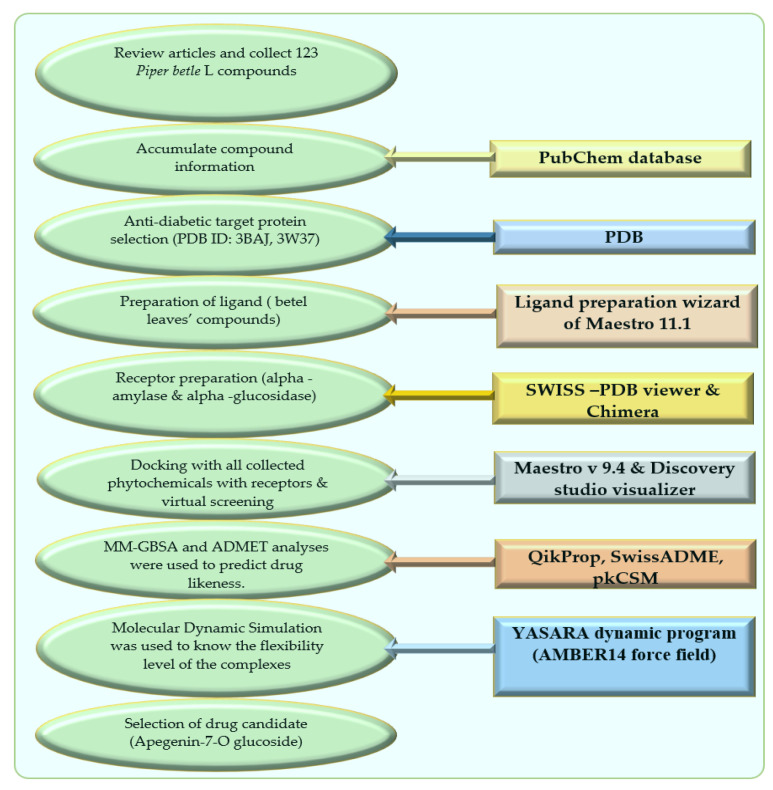
Schematic diagram of this study with molecular docking and dynamics simulation of natural compounds from *Piper betle* L.

**Table 1 molecules-27-04526-t001:** Data for the molecular docking of apigenin-7-*O*-glucoside, luteolin-7-*O*-glucoside, quercetin, and control acarbose with alpha-amylase (3BAJ).

Compounds	Interaction	Residues in Contact	Distance in Å
Apigenin-7-*O*-glucoside	Conventional hydrogen bond	ASP-300	2.41
		GLU-233	1.87
		ASP-197	1.86, 1.59
		HIS-305	2.65
		GLY-306	2.89
		GLN-63	2.87
	Carbon hydrogen bond	ASP-300	2.44
	Unfavourable donor-donor	ARG-195	2.42
	Pi-Pi stacked	TRP-59	
Luteolin-7-*O*-glucoside	Conventional hydrogen bond	ASP-300	1.68
		GLU-233	1.97
		HIS-299	2.35
		ASP-356	2.26, 2.82
		ARG-195	2.16
		GLN-63	2.27, 2.73, 2.89
	Carbon hydrogen bond	ASP-197	2.37
		ASP-300	2.23
		HIS-305	2.54
	Pi-cation	HIS-305	2.52
	Pi-Pi stacked	TRP-95	4.92, 5.55
			4.03, 4.29
Quercetin	Conventional hydrogen bond	ASP-300	2.22
		ASP-197	1.78
		HIS-305	2.19, 2.84
		THR-163	2.09
	Carbon hydrogen bond	HIS-101	2.4
Acarbose	Conventional hydrogen bond	GLU-240	2.20, 2.02
		GLY-306	1.99, 1.73
		HIS-305	2.93, 2.11
		ASP-197	1.81
		ASP-300	1.68
		THR-163	3.01, 2.19
	Carbon hydrogen bond	GLY-306	2.54
		ASP-300	2.51
	Pi-Pi stacked	TYR-151	3.84

**Table 2 molecules-27-04526-t002:** Data for the molecular docking of apigenin-7-*O*-glucoside, luteolin-7-*O*-glucoside, quercetin, and control acarbose with alpha-glucosidase (3W37).

Compounds	Interaction	Residues in Contact	Distance in Å
Apigenin-7-*O*-glucoside	Conventional hydrogen bond	ASP-60	1.7
		ASN-258	2.01
		ASP-327	2.27, 2.98
		ILE-143	1.75, 2.61
		ASP-382	1.77, 2.06
	Carbon	ARG-411	2.05
		GLY-384	3.09
		GLY-410	2.67
	Pi-Anion	ASP-327	3.98, 4.56
	Pi-Pi stacked	PHE-163	4.52
	Pi-Pi T shaped	TYR-63	5.44
Luteolin-7-*O*-glucoside	Conventional hydrogen bond	HIS-103	2.96
		ASP-60	1.61, 1.80
		ILE-143	1.75, 2.58
		ASP-382	1.77, 2.07
		THR-409	2.45
		ASN-258	2.04
		ARG-411	1.79, 2.05
	Carbon hydrogen bond	ASP-327	2.93, 3.99
		GLY-410	2.67
		GLY-384	3.09
	Pi-Anion	ASP-199	4.32
	Pi-Alkyl	ALA-200	5.24
	Pi-Pi T shaped	PHE-144	5.8
	Pi-Pi stacked	PHE-163	4.26
Quercetin	Conventional hydrogen bond	HIS-203	2.07
		ASN-258	2.08, 2.89, 2.90
		ASP-382	2.02, 2.07
	Pi-cation	ARG-411	4.97
	Pi-Pi T-shaped	PHE-163	5.13
	Pi-Alkyl	ILE-143	5.04, 5.16
Acarbose	Conventional hydrogen bond	ASP-327	190
		ARG-411	2.56, 2.02
		ASP-60	1.48
		GLN-167	3.1
		HIS-103	2.78
		ASP-199	1.60, 1.92
		HIS-203	2.07, 1.89, 2.31
		GLY-384	1.95
		SER-145	2.14
	Carbon hydrogen bond	ASP-60	2.26

**Table 3 molecules-27-04526-t003:** Pharmacokinetic and toxicological properties of apigenin-7-*O*-glucoside calculated from QikProp.

Compound	Pubchem Id	Docking Score	MMGBSA Dg Bind *	Molecular Weight (MW) ^a^	SASA ^b^	Donor HB ^c^	Accept HB ^d^	Qplog Po/w ^e^	QPlogS ^f^	QPlog HERG ^g^	Human Oral ^h^
Apigenin-7-*O*-glucoside for amylase	5280704	−7.6	−45.02	432.4	680.5	5	12.25	−0.307	−3.248	−5.79	30.65
Apigenin-7-*O*-glucoside for glucosidase	5280704	−10.2	−38.28	432.4	680.5	5	12.25	−0.307	−3.248	−5.79	30.65

* MM-GBSA, Molecular mechanics-generalized born and surface area; ^a^ Molecular weight (acceptable range: <500); ^b^ Total solvent accessible surface area in using a probe with a 1.4 radius (acceptable range: 300–1000); ^c^ Hydrogen bond donor (acceptable range: ≤5); ^d^ Hydrogen bond acceptor (acceptable range: ≤10); ^e^ Predicted octanol/water partition coefficient (acceptable range: −2 to 6.5); ^f^ Predicted aqueous solubility, S in mol dm^−3^ (acceptable range: −6.5 to 0.5); ^g^ Predicted IC50 value for blockage of HERG K+ channels (concern: below −5); ^h^ Predicted human oral absorption on 0 to 100% scale (<25% is poor and >80% is high).

## Data Availability

Data are available in this study. If additional data are required upon request to the corresponding author.

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
