# Peer review of "Molecular Docking and Dynamics Simulation of Natural Compounds from Betel Leaves (Piper betle L.) for Investigating the Potential Inhibition of Alpha-Amylase and Alpha-Glucosidase of Type 2 Diabetes"

_molecules, 2022, doi:10.3390/molecules27144526_

Round 1
Reviewer 1 Report
In manuscript entitled “Molecular Docking and Dynamics Simulation of Natural Com-2 pounds from Betel leaves (Piper betle L.) for Investigating the 3 Potential Inhibition of Alpha-amylase and Alpha-glucosidase 4 of Type 2 Diabetes” authors presented molecular docking studied and molecular dynamics simulation of selected molecules toward Alpha-amylase and Alpha-glucosidase 4 with the purpose to assess selected molecules as inhibitors of stated enzymes and potential therapeutics for Type 2 Diabetes. Authors selected molecules from Betel leaves (Piper betle L.), well establish medicinal plant. In my opinion authors selected appropriate molecules and enzymes for establishing potential therapeutics. I find that this approach has significant scientific novelty in the field of computer aided drug design. Methodology used for stated aims is appropriate and, in my opinion, there is no need to perform additional calculations or to include additional methods. Results are clearly presented and conclusions are consistent with the evidence and arguments presented so they address completely the main aim of this research. Authors cited appropriate references. To conclude, this is well written paper that presents interesting and well performed research. I recommend publication after regular language, text and technical editing.
Author Response
Comment 1: Accepted
Response: Thank you for your approval.
Comment 2: “I recommend publication after regular language, text and technical editing”.
Response 2: The manuscript was updated according to reviewer comments.
Reviewer 2 Report
Dear Authors:
In this paper, the authors conducted a virtual screening of compounds from Piper betle against alpha-amylase and alpha-glucosidase to obtain a lead candidate. Results look promising, but further discussion is needed.
The methods used are standard, but for validation, docking and MM-GBSA calculations for alpha-amylase and alpha-glucosidase should be performed for known compounds to show reproducibility of docking poses and scoring.
For each of alpha-amylase and alpha-glucosidase, show a 3D view aligning the docking pose of Apigenin-7-O-glucoside with the X-ray pose of acarbose. The validity of the docking poses of Apigenin-7-O-glucoside should then be discussed based on the similarity of the ligand orientation and interaction.
Correct the following points as well.
In 2.3. Ligand Binding Analysis, it is difficult to understand whether each paragraph is describing alpha-amylase or alpha-glucosidase.
In P4L178-179, figure 2. A-b -> figure 2. a-b
In Figure 1 and 2, the docking poses of luteolin-7-O-glucoside should also be shown.
In Table 2 and 3, for comparison, the interactions of acarbose should also be shown.
In P14L510, less (<1Å) RMSD, what is this RMSD with?
In References, the uses of et al. are incorrect.
Author Response
Comment 1: Results look promising, but further discussion is needed.
Response 1: Thanks for your excellent suggestion, according to reviewer comments we have improved major parts of the discussion section (Line no 326-329, 344-352, 366-376, 412-417, 433-440) and added new references (ref. NO 4, 37, 46, 47, 48).
Comment 2: “The methods used are standard, but for validation, docking and MM-GBSA calculations for alpha-amylase and alpha-glucosidase should be performed for known compounds to show reproducibility of docking poses and scoring”.
Response 2: Thanks for your nice comment. We already showed the control compound (acarbose, an established anti-diabetic drug which is used to treat diabetes mellitus type 2). Based on MM-GBSA calculations for alpha-amylase and alpha-glucosidase, we found three ligands based on the highest binding affinity (ΔG Bind) against both alpha-amylase and alpha-glucosidase. Among them, Apigenin-7-O-glucoside showed the highest affinity (ΔG Bind = -45.02 kcal mol-1 against alpha-amylase, and -38.28 kcal mol-1 against alpha-glucosidase). Compared to control inhibitor acarbose (-36.796 kcal mol-1 for alpha-amylase and -29.622 kcal mol-1 for alpha-glucosidase), we showed the Apigenin-7-O-glucoside as the best compound, which is ultimately used for further research of drug candidate (see the result section 2.2)
Comment 3: “For each of alpha-amylase and alpha-glucosidase, show a 3D view aligning the docking pose of Apigenin-7-O-glucoside with the X-ray pose of acarbose. The validity of the docking poses of Apigenin-7-O-glucoside should then be discussed based on the similarity of the ligand orientation and interaction”
Response 3: Thanks for the recommendation. However, before docking with the Piper betle compound, we redocked the established and native ligand acarbose with alpha amylase and alpha glucosidase (added as supplementary figure). As the ligand posses several heteroatoms thus, there was a slight RMSD difference between x-ray co-crystal acarbose and redocked acarbose. However, the RMSD difference was around 2.25 Å (for alpha amylase) and 2.6 Å (for alpha glucosidase) which lies in the acceptable range (Ramírez D et al. Molecules. 2018. PMC6102569). Similar residual interactions denote the effectivity of selected compounds.
Comment 4: “In 2.3. Ligand Binding Analysis, it is difficult to understand whether each paragraph is describing alpha-amylase or alpha-glucosidase”
Response 4: We re-organize and rewrite the result section as well as 2.3 section (Ligand Binding Analysis). In this section, we describe first alpha-amylase with Apigenin-7-O-glucoside, then alpha-glucosidase with Apigenin-7-O-glucoside and control acarbose interaction with alpha-amylase and alpha-glucosidase.
Comment 5: “In P4L178-179, figure 2. A-b -> figure 2. a-b
Response 5: Updated
Comment 6: “In Figure 1 and 2, the docking poses of luteolin-7-O-glucoside should also be shown.
Response 6: We added a new figure 3, docking poses of luteolin-7-O-glucoside with alpha-amylase and alpha-glucosidase.
Comment: 7: In Table 2 and 3, for comparison, the interactions of acarbose should also be shown.
Response 7 : Thanks for a nice suggestion . We added the control acarbose interaction in Table 2 and Table 3.
Comment 8: “In P14L510, less (<1Å) RMSD, what is this RMSD with?”
Response 8: The RMSD value less than 1 angstrom provide a message of conformation with reference (acarbose) and target proteins (alpha-amylase and alpha-glucosidase). The RMSD value gives the average deviation between the corresponding atoms of two proteins: The smaller the RMSD like the structure and orientation.
Comment 9: “In References, the uses of et al. are incorrect”.
Response 9: All references are fixed in the reference section as well as in the manuscript text.
Reviewer 3 Report
In this work, the authors studied the inhibitory action of α-amylase and α- glycosidase against type 2 diabetes of the compounds of P. betle L. through molecular docking, molecular dynamics simulation, and ADMET analysis. The results indicates the apigenin-7-O-glucoside was exposed to be the most stable molecule with the highest binding free energy through molecular dynamics simulation, indicating that it could compete with the inhibitors' native ligand. This work is interesting, and the data is rich. Therefore, the manuscript can be accepted after a minor revision.
Fig.4 is not clear. Please revise.
Some references are cited irregularly, such as Ref.63.
The BSSE correction should be considered when calculated the binding energy.
Author Response
Reviewer #3:
Comment 1. “the manuscript can be accepted after a minor revision”.
Response 1: Thanks for your positive impression and acceptance
Comment 2. Fig.4 is not clear. Please revise.
Response 2: Updated and the precise figure is provided
Comment 3: Some references are cited irregularly, such as Ref.63.
Response 3: Irregular references are deleted, and updated the all references
Comment 4: The BSSE correction should be considered when calculated the binding energy.
Response 4: Thanks for your nice comment and suggestion. However, we try to consider the Basis Set Superposition Error (BSSE) correction.
Reviewer 4 Report
The strategy is interesting; however, the manuscript must be restructured by ordering the main ideas.
The statement on line 109 is inappropriate for the section.
The way of presenting the results must be reorganized, considering that the central objective of the study is the applicability of in-silico studies focused on selecting molecules with greater inhibitory potential of the selected enzymes.
Modify the wording of the in-silico studies, establishing a chronological order of the findings that made it possible to identify apigenin-7-O-glucoside as the best candidate to continue with pharmacological studies.
At the end of the results, describe the prediction studies of the pharmacokinetic characteristics of the molecules under study, as well as their potential toxic effect.
The discussion must be restructured, eliminating from the manuscript the pharmacological effects of other plant species and secondary metabolites that have no structural relationship with the molecules selected in the study as candidates. For this, it is suggested to review and include the information reported in:
Comparative Study of Dietary Flavonoids with Different Structures as α-Glucosidase Inhibitors and Insulin Sensitizers. J Agric Food Chem. 2019 Sep 18;67(37):10521-10533. doi: 10.1021/acs.jafc.9b04943. Epub 2019 Sep 10.
Comparative Study of Dietary Flavonoids with Different Structures as α-Glucosidase Inhibitors and Insulin Sensitizers. Sci Rep 2018 Apr 3;8(1):5471. doi: 10.1038/s41598-018-23736-1.
Line 361 indicates that alpha-glucosidases are pancreatic enzymes, a statement that is incorrect.

Author Response
Reviewer #4:
Comment 1: The strategy is interesting; however, the manuscript must be restructured by ordering the main ideas.
Response: Thanks for your suggestions. The whole manuscript restructure is based on all comments from all reviewers.
Comment 2:” The statement on line 109 is inappropriate for the section.
Response 2: Updated the sentence and rewrite the 109-112 no line
Comment 3 : “The way of presenting the results must be reorganized, considering that the central objective of the study is the applicability of in-silico studies focused on selecting molecules with greater inhibitory potential of the selected enzymes.
Response 3: Thanks for your valuable comments on improving and re-organized the results. We update or rewrite the study’s main objective in the last section of introduction (109-112 no line).
Comment 4: “Modify the wording of the in-silico studies, (computational approaches) establishing a chronological order of the findings that made it possible to identify apigenin-7-O-glucoside as the best candidate to continue with pharmacological studies.
Response 4: We modify the wording of in-silico studies, we arranged or organized the complete manuscript as reviewer comments (we describe as recommends of reviewer with chronological order of the findings that made it possible to identify apigenin-7-O-glucoside as the best candidate)
Comments 5: At the end of the results, describe the prediction studies of the pharmacokinetic characteristics of the molecules under study, as well as their potential toxic effect.
Response 5: We added separate paragraph in section 2.5 ( Pharmacokinetics and drug likeliness properties) and added a supplementary table S1.
Comments: “The discussion must be restructured, eliminating from the manuscript the pharmacological effects of other plant species and secondary metabolites that have no structural relationship with the molecules selected in the study as candidates. For this, it is suggested to review and include the information reported in:”
Response: We deleted other plant species except betel leaves-related research and updated the discussion section of this study.
Comments: “Line 361 indicates that alpha-glucosidases are pancreatic enzymes, a statement that is incorrect.
Response: fixed the sentence
Round 2
Reviewer 2 Report
Dear Authors:
I recommend publication of this manuscript in molecules because it has been appropriately revised.
However, the following corrections should be made for publication.
Figure 1 and 2: Residue labels are illegible.
P4L188: Alpha-glucosidase -> alpha-glucosidase
In the text, XAAnnn seems to be a residue in alpha-amylase and XAA-nnn be a residue in alpha-glucosidase, but this should be annotated somewhere. Also, some of them, such as Asp 300, do not follow the rule, so they should be unified.
Author Response
Comment: “I recommend publication of this manuscript in molecules because it has been appropriately revised.
Response: Thanks for your positive response and nice suggestions.
Comment: “Figure 1 and 2: Residue labels are illegible.
Response: Update the figure 1 and 2 and fixed unwanted mistakes
Comment: “P4L188: Alpha-glucosidase -> alpha-glucosidase”
Response: Fixed
Comment: “In the text, XAAnnn seems to be a residue in alpha-amylase and XAA-nnn be a residue in alpha-glucosidase, but this should be annotated somewhere. Also, some of them, such as Asp 300, do not follow the rule, so they should be unified.”
Response: Carefully checked and updated.
Reviewer 4 Report
Substantial changes were made in the description of results and discussion. The manuscript is understandable and presents a contribution to the in silico study of molecules with potential pharmacological application.
Author Response
Comment: Substantial changes were made in the description of results and discussion. The manuscript is understandable and presents a contribution to the in silico study of molecules with potential pharmacological application.
Response: Thanks for your nice comments and suggestions.